# *Cis*-Allosteric Regulation of HIV-1 Reverse Transcriptase by Integrase

**DOI:** 10.3390/v15010031

**Published:** 2022-12-21

**Authors:** Takao Masuda, Osamu Kotani, Masaru Yokoyama, Yuya Abe, Gota Kawai, Hironori Sato

**Affiliations:** 1Department of Immunotherapeutics, Graduate School of Medical and Dental Sciences, Tokyo Medical and Dental University (TMDU), Yushima, 1-5-45 Bunkyo-ku, Tokyo 113-8519, Japan; 2Laboratory of Viral Genomics, Pathogen Genomics Center, National Institute of Infectious Diseases, Gakuen, 4-7-1, Musashimurayama-shi, Tokyo 208-0011, Japan; 3Department of Life Science, Faculty of Advanced Engineering, Chiba Institute of Technology, 2-17-1 Tsudanuma, Narashino-shi, Chiba 275-0016, Japan

**Keywords:** HIV-1, reverse transcriptase, integration, integrase, pol, protease, deoxyribonucleoside triphosphates, molecular dynamics

## Abstract

Reverse transcriptase (RT) and integrase (IN) are encoded tandemly in the *pol* genes of retroviruses. We reported recently that HIV-1 RT and IN need to be supplied as the *pol* precursor intermediates, in which RT and IN are in fusion form (RTIN) to exert efficient reverse transcription in the context of HIV-1 replication. The mechanism underlying RTIN’s effect, however, remains to be elucidated. In this study, we examined the effect of IN fusion on RT during reverse transcription by an in vitro cell-free assay, using recombinant HIV-1 RTIN (rRTIN). We found that, compared to recombinant RT (rRT), rRTIN generated significantly higher cDNAs under physiological concentrations of dNTPs (less than 10 μM), suggesting increased affinity of RTIN to dNTPs. Importantly, the cleavage of RTIN with HIV-1 protease reduced cDNA levels at a low dose of dNTPs. Similarly, sensitivities against RT inhibitors were significantly altered in RTIN form. Finally, analysis of molecular dynamics simulations of RT and RTIN suggested that IN can influence the structural dynamics of the RT active center and the inhibitor binding pockets in *cis*. Thus, we demonstrated, for the first time, the *cis*-allosteric regulatory roles of IN in RT structure and enzymatic activity.

## 1. Introduction

Reverse transcriptase (RT) and integrase (IN) are essential enzymes encoded in the *pol* gene for retroviruses to establish proviral DNA, in which viral genomic RNA (gRNA) is reverse transcribed into DNA followed by its integration into the host cell chromosome [1]. Reverse transcription and the integration of a virus genome are catalyzed by RT [1] and IN [2], respectively. Human immunodeficiency virus type 1 (HIV-1) infection depends on the activation status of target cells, CD4+ T cells [3] and monocytes/macrophages [4]. SAMHD1 has been identified as a host restriction factor against HIV-1 infection by reducing cellular levels of deoxyribonucleoside triphosphates (dNTPs) [5]. Reverse transcription is therefore a critical step depending on cellular status, especially on the concentration of dNTPs.

Retrovirus RT contains two independent catalytic sites for RNA-dependent DNA polymerase (RDDP) and RNase H [6] activities to generate DNA copies of vRNA [1]. IN has one catalytic site for 3′-end processing and a strand-transfer of viral DNA ends to integrate viral DNA into the host chromosome [2]. Meanwhile, the additional roles of IN at steps prior to integration, including viral maturation and reverse transcription, have been reported for HIV-1 [4,7,8,9,10,11,12], which has become a target for the development of a new class of IN inhibitors known as allosteric inhibitors (ALLINIs) [13,14]. Interaction of IN with virus genomic RNA (vRNA) has been reported to be crucial for HIV-1 virion morphogenesis [15,16,17], providing a molecular base for pleiotropic effects of IN mutations and/or ALLINIs [12]. 

HIV-1 RT and IN are originally expressed as parts of Gag-Pol precursor polyproteins for assembly into viral particles, followed by protease-mediated cleavage into individual RT and IN during or after virus release [18,19]. Our previous *trans*-complementation assay in the context of virus replication showed that HIV-1 IN might exert a crucial role during reverse transcription through the Pol precursor intermediate, in which RT and IN were kept in fusion form (RTIN) [20]. However, the mechanism underlying the critical role(s) of RTIN during reverse transcription remains to be elucidated. 

In this study, to delineate the impact of HIV-1 IN fusion to RT during reverse transcription, we prepared recombinant RTIN protein (rRTIN) and evaluated the impact of HIV-1 IN fusion to RT during reverse transcription using an in vitro assay [21,22]. We also conducted MD simulations of RT and RTIN proteins. The results provided direct evidence of a novel *cis*-regulatory role of IN during HIV-1 reverse transcription through the RTIN fusion form.

## 2. Materials and Methods

### 2.1. Construction of Expression Vectors

For His-tagged recombinant protein expression in the *E. coli*, the entire RTIN region of HIV-1 was amplified with primer sets of ^5′^GGA CCC GGG CCC ATT AGT CCT ATT GAG ACT GTAC^3′^ and ^5′^AGC CTC GAG TTA ATC CTC ATC CTG TCT ACT^3′^ using a pNL4-3lucΔenv vector [8] as a template. Fragments of RT carrying IN NTD (residues 1-55) or NTD and CCD (residues 1-212) were similarly generated, using primer sets with a reverse primer of ^5′^AGC CTC GAG TTA GTC TAC TTG TCC ATG CAT or ^5′^AGC CTC GAG TTA TTC TTT AGT TTG TAT GTC, respectively. The amplified fragment was cloned into a pET-47b (+) vector (Novagen, Madison, WI, USA) at Sma I and Xho I sites. Entire RTIN regions were verified by DNA sequence analysis.

### 2.2. Preparation of Recombinant RTs 

A His-tagged form of HIV-1 RTIN or RT(p66) was expressed in the *E. coli* BL21 derivative strain of Rosetta (DE3) (Invitrogen/Thermo Fisher Scientific, Waltham, MA, USA) by incubation with 0.1 mM IPTG for 16 hr at 18 °C. Cells were incubated in lysis buffer (BugBuster^®^ Protein Extraction Reagent; Merck, Darmstadt, Germany) with lysosome (FUJIFILM-Wako, Osaka, Japan) and nuclease (Benzonase, Merck/Millipore, Burlington, MA, USA) for 1 hr. After cell lysis, the soluble fraction was obtained by centrifugation at 10,000× *g* for 30 min. His-tagged RTIN or RT in the soluble fraction was purified through a Ni-affinity column (Ni-NTA agarose, Qiagen, Hilden, Germany). The His-tag was then removed by treatment with HRV3C protease (Turbo3C, Nacalai-Tesque, Kyoto, Japan), followed by glutathione-Sepharose column purification (Glutathione Sepharose™ 4B, Cytiva Life Science, Marlborough, MA, USA) to remove the HRV3C. The flow-through fraction was then subjected to cation-exchange chromatography (SP-HP, GE Healthcare, Chicago, IL, USA). Fractions eluted by SP buffer [20 mM HEPES-NaOH (pH 7.0), 0.1 mM EDTA, 0.01% TritonX-100, 10% glycerol, and 1 mM DTT] containing 250–500 mM NaCl were pooled. The purified recombinant HIV-1 RT IN (rRTIN) was concentrated through a concentrator column with a filtration cutoff of 50 kDa (VIVAspin 50K, GE Healthcare) and stored in SP buffer containing 300 mM NaCl at −80 °C. To estimate the purity and/or oligomer status of rRTIN, an aliquot of purified rRTIN was subjected to size-exclusion chromatography (SEC) analysis using a Superdex 75 or 200 Increase 100/300 GL column operated by the AKTA go system (Cytiva Life Science, Marlborough, MA, USA). The elution profile was analyzed by UNICORN ver. 7.4 software (Cytiva Life Science, Marlborough, MA, USA). SEC fractions were subjected to SDS-PAGE followed by staining with Coomassie brilliant blue.

### 2.3. Proteolysis of RTIN by HIV-1 Protease

Purified rRTIN was treated with HIV-1 protease (PROSPEC, Rehovot, Israel) in PR-buffer [20 mM Tris-HCl (pH7.0) and 1 M NaCl] for 30 min at 37 °C. Proteolysis of rRTIN was verified by SDS-PAGE analysis and Western blot analyses using anti-HIV-1 RT or an IN antibody (Abcam, Cambridge, UK). 

### 2.4. In Vitro Reverse Transcription Assay

An in vitro reverse transcription assay was performed as described. Briefly, 100 ng of synthetic HIV-1 RNA (corresponding to 0.1 pmole) and 10 pmole of pbs-sRNA primers (Sigma-Aldrich Japan Genosis, Tokyo, Japan) were annealed by heating at 70 °C for 10 min, followed by cooling on ice. Then, vRNA/pbs-sRNA mixtures were incubated with 20 pmoles of synthetic NCs (Peptide Institute, Osaka, Japan) for 15 min. A serial dilution of vRNA/pbs-sRNA/NC was then inoculated in the reaction buffer [50 mM Tris-HCl (pH8.3), 75 mM KCl, 3 mM MgCl_2_, 10 mM DTT, 0.1–100 μM dNTPs and 8 pmol of rRT or rRTIN]. The reaction was initiated by incubation at 42 °C to avoid mis-annealing of vRNA to facilitate efficient cDNA synthesis and correct 1st strand-transfer as described before [22]. At 30–300 min, the reaction was terminated by heating at 75 °C for 10 min. Following dilution with TE buffer (10 mM Tris-HCl [pH8.0], 1 mM EDTA), the number of cDNA spices was determined by qPCR assay using primer sets specific to the R/u5, U3/u5 or U3/pbs region. For velocity analysis, the copy number of cDNAs generated during 60 min incubation was determined by q-PCR using the R/u5 primer set. The efficiency of f strand transfer of -sscDNA (1st strand-transfer) was estimated by calculating the amount of the U3/u5 relative to that of R/u5 products as described previously [22].

### 2.5. Reagents

Efavirenz and Nevirapine (Tokyo Chemical Industry, Tokyo, Japan) were resolved in dimethyl sulfoxide (DMSO). 3’-Azido-2’,3’-dideoxythymidine-5’-triphosphate (AZT-TP, Jena Bioscience, Jena, Germany) was resolved in nuclease-free distilled water. Raltegravir (Cayman Chemical, Ann Arbor, MI, USA) was resolved in DMSO.

### 2.6. In Silico Analysis of HIV-1 RT and RTIN

Molecular dynamics (MD) simulations: A three-dimensional model of RTIN of HIV-1 was constructed by the AlphaFold 2 program [23] using the reported amino acid sequence of the HIV-1 NL4-3 infectious molecular clone (GenBank accession no. AAK08484) [24]. The obtained RTIN model gave a per-residue confidence score (pLDDT) of 83.3, which is in the “Confident” range [23]. An RT model was constructed from the RTIN model by deleting the IN region using the Molecular Operating Environment (MOE) (Chemical Computing Group, Montreal, Quebec, Canada). The accuracy of the RTIN model was evaluated by the root-mean-square deviation (RMSD) score between the predicted RTIN model and the reported X-ray crystal structure of the RT or IN (PDB code: RT; 3HVT [25], IN; 1K6Y [26] and 1EX4 [27]) using the “Structure Superposition” tool in MOE. Each domain of RTIN model was similar to the reported RT or IN models (RMSD score: RT; 2.161 Å, IN-NTD; 1.931 Å, IN-CCD; 1.81 Å, IN-CTD; 0.785 Å). The RTIN and RT models were subjected to MD simulations as described for HIV-1 capsid protein [28] and other viral proteins [29,30,31]. Briefly, the simulations were performed using the pmemd.cuda.MPI module in the Amber 16 program [32] with the ff14SB force field for protein simulation [33]. The RTIN and RT models were solvated in a truncated octahedral box of TIP3P water molecules with a distance of at least 9 Å around the models [34]. A non-bonded cut-off of 10 Å was used. Bond lengths involving hydrogen were constrained with SHAKE, a constraint algorithm that satisfies Newtonian motion [35]. The time step for all MD simulations was set to 2 fs. After heating calculations were performed for 20 ps up to 310 K using the NVT ensemble, simulations were executed using the NPT ensemble at 1 atm, at 310 K and in 150 mM NaCl for a total of 500 ns.

Root-mean-square deviation (RMSD): The trajectory files during MD simulations were used to calculate RMSD. RMSDs between the heavy atoms of the initial complex structure and the structure at given time points during the MD simulation were calculated to monitor the overall structural changes as described previously [29,30,31]. Calculations of RMSDs were done by the *cpptraj* module in AmberTools 16, a trajectory analysis tool [32].

Molecular surface area: The trajectory files during MD simulations were used to calculate the molecular surface area of the RT and RTIN using the linear combinations of pairwise overlaps algorithm [36] in *cpptraj* operated in AmberTools 16 [32].

Hydrogen bond: A hydrogen bond in a protein was identified by geometric criteria, in which the bond is defined by the distance and angle between an acceptor heavy atom and a donor heavy atom in a given amino acid pair. The trajectory files during MD simulations were used to calculate the number of hydrogen bonds between the fingers and thumb subdomains in the RT region using the *cpptraj* module in AmberTools 16 [32].

Root-mean-square fluctuation (RMSF): The trajectory files during the last 100 ns of MD simulations (*n* = 50,000) were used to calculate RMSF. The RMSF of the Cα atoms of amino acid residues was calculated to obtain information about the atomic fluctuations of individual amino acid residues of RTIN protein during MD simulations, using the *cpptraj* module in AmberTools 16 [32].

## 3. Results

### 3.1. Preparation of Recombinant RTIN

To evaluate the direct contributions of IN through its fusion with RT (RTIN) during reverse transcription in vitro, recombinant HIV-1 RTIN (rRTIN) was prepared by using an *E. coli* expression system based on the protocol for recombinant RT (rRT) [22] (Figure 1a). However, we found that a majority of rRTIN was in insoluble form and precipitated in the inclusion body during the induction protocol with 1 mM IPTG at 37 °C, the protocol for the rRT preparation. To reduce the aggregation of rRTIN in the inclusion body, mild induction to express rRTIN was carried out with 0.1 mM IPTG at 18 °C for 20 hrs. Using this modified induction protocol, we successfully obtained rRTIN in soluble form. Size-exclusion column (SEC) analysis and SDS-PAGE analysis revealed that a majority (~80%) of the rRTIN may have been formed from a hexamer or a higher multimer (Appendix A). Moreover, the proteolysis profile of rRTIN by HIV-1 protease was examined. As expected, HIV-1 protease cleaved rRTIN into RT subunits (p66 and p51) and IN (p32), which were reacted to anti-RT or IN antibodies, respectively (Figure 1b), thus verifying the successful preparation of the rRTIN. The faint bands for p66 and p51 detected in rRTIN preparation without HIV-1 protease (Figure 1b, Prot-) may be generated most probably by proteolysis with *E. coli*-derived protease(s) during the induction phase of rRTIN. 

### 3.2. Evaluation of rRTIN Enzymatic Activities during Reverse Transcription

The enzymatic activity of rRTIN was addressed by an in vitro reverse transcription assay as described previously [21,22]. The levels of cDNA intermediates were monitored by q-PCR using a specific primer set to amplify the R/u5, U3/u5, or U3/pbs region of the HIV-1 genome. Amplified products of R/u5, U3/u5, or U3/pbs reflect cDNA intermediates of minus-strand strong-stop cDNA (−sscDNA), strand-transfer products of -sscDNA (1st strand-transfer), or plus-strand strong-stop cDNA (+sscDNA), respectively [22]. Levels of these cDNA products could be used to monitor the RT enzyme activities of RNA-dependent DNA polymerase (RDDP), RNase H, and DNA-dependent DNA (DDDP) polymerase activities (Figure 2a). The level of -sscDNA products (R/u5) was efficiently generated and accumulated linearly from 30 min to 60 min of incubation, followed by sequential accumulation of the 1st strand-transfer products (U3/u5) and +sscDNA products (U3/pbs) (Figure 2b). These results demonstrated that purified rRTIN possesses all RDDP, RNase H, and DDDP activities.

### 3.3. Impact of IN Fusion on RT Functions

We directly compared RT activities of rRTIN and rRT in modified assay conditions in which the amounts of the reaction substrates were varied to address differences between these proteins in detail. As the critical role of IN during HIV-1 virion morphogenesis through interaction with vRNA has been reported [15,16,17], we first compared -sscDNA (R/u5) levels generated by rRTIN or rRT under different concentrations of synthetic vRNA [22] (Figure 3a). In this experiment, the velocity of -sscDNA synthesis during 60 min of the reaction was measured; data are shown as values relative to the velocity at 100 ng vRNA as 1.0. From the velocity plot results, we estimated the affinity of rRTIN or rRT for vRNA (Km for vRNA) by calculating the vRNA concentration to give a relative velocity of 0.5. The estimated Km value for vRNA was not significantly different between rRTIN and rRT (Figure 3b). We also found that rRTIN possessed the first strand-transfer (Figure 3c) and +sscDNA synthesis (Figure 3d) efficiencies equivalent to those of rRT. These results demonstrated that rRTIN possessed RDDP, RNase H and DDDP activities that were almost equivalent to those of RTp66 under these experimental conditions.

### 3.4. Impact of IN Fusion on Affinity of RT to dNTPs

Cellular levels of deoxyribonucleoside triphosphates (dNTPs) are critical determinants for retrovirus reverse transcription and infection [5]. We next examined -sscDNA generation under different concentration of dNTPs with a fixed amount of vRNA (100 ng per reaction). The velocity of R/u5 (-sscDNA) synthesis during 60 min of the reaction was measured, and data are shown as values relative to the velocity at 100 μM dNTPs as 1.0. Here, we noticed that rRTIN produced R/u5 products with significantly higher efficiency at low doses of dNTPs (1 or 10 μM) compared with rRTp66 (Figure 4a). We also examined a heterodimer form of rRT (p66/p51) in parallel. From the velocity plot results, we estimated the Km value of rRTIN or rRTs for dNTPs by calculating a concentration of dNTPs to give a relative velocity of 0.5. It should be noted that the Km value of rRTIN was more than ~6-fold lower than that of rRTp66 or rRTp66/p51, suggesting that rRTIN possesses a significantly higher affinity to dNTPs than rRTs (Figure 4b).

To address the crucial role of IN fusion to RT, the effect of proteolytic cleavage of RTIN by HIV-1 protease was addressed. Since HIV-1 protease produced a heterodimeric form of RT composed of p66 and p51subunits, rRT in the heterodimeric form (p66/p51) was also examined in parallel. When rRTIN was treated with HIV-1 protease, efficient synthesis of -sscDNA at low doses of dNTPs (1 or 10 μM) was severely abrogated (Figure 4c). The estimated Km values for the dNTPs were increased significantly, to the level of rRT (p66/p51), by protease treatment of RTIN (Figure 4d). These data demonstrated that the fusion of IN to RT is critical to the exertion of a stimulatory effect on RDDP activity at lower doses of dNTPs.

### 3.5. Effects of RTIN on RT and IN Inhibitors

The experiments above demonstrated that RT RDDP activity is stimulated in RTIN form, most probably by inducing a conformational change of RT that might confer increased affinity to dNTPs. We next examined the drug sensitivity of RTIN against an RT or IN inhibitor by measuring -sscDNA generation under different concentrations of efavirenz (EFV) or raltegravir (RAL). The levels of R/u5 products were measured and are shown as values relative to the level with solvent (DMSO) only as 1.0 (Figure 5a). In comparison with RTp66, we found that RTIN showed lower sensitivity to inhibition by EFV. At 1 nM of EFV, where more than 90% of RDDP was inhibited for RTp66, no apparent inhibitory effect was detected for RTIN. The effective concentration 50 (EC_50_) of EFV to RTIN or RTp66 was calculated to be 12.0 ± 3.16 nM or 0.507 ± 0.01 nM, respectively (Figure 5b). This result demonstrated that the drug sensitivity of EFV on RTIN was reduced significantly, by 23.7-fold, compared to RTp66. In contrast, no apparent inhibition by RAL was detected for RTIN and RTp66 up to 100 μM, indicating that the catalytic activity of IN might not be involved in the RDDP activity of RTIN or insufficient interaction of RAL with RTIN. We also examined another non-nucleoside RT inhibitor, nevirapine (NEV), and a nucleoside RT inhibitor, 3’-Azido-2’,3’-dideoxythymidine-5’-triphosphate (AZT-TP). The inhibitory effects of NEV and AZT-TP on RTIN or RTp66 were similarly examined (Figure 5b). Although the magnitudes of the effect of RTIN compared to RTp66 were diverse, RTIN was again found to significantly affect the EC_50_ of EFV and AZT-TP. The differences in the degrees of effect among RTIs reflect that the RT conformational change introduced by IN fusion might induce local changes near the RT catalytic center. As noticed regarding the effect of RTIN on affinity to dNTPs (Figure 4), we observed that protease treatment of rRTIN significantly increased sensitivity to EFV (Figure 5c), demonstrating, here again, an allosteric regulation of HIV-1 RT conformation through RTIN form. 

HIV-1 IN contains three structural domains: NTD, CCD, and CTD [37]. To address the contribution of IN domains to the stimulatory effects of RDDP activity, rRT carrying IN NTD (residues 1–55, rRTIN_55_) or NTD and CCD (residues 1–212, rRTIN_212_) were prepared (Appendix A) and their RDDP activity levels were evaluated under different concentrations of dNTPs (Appendix A). Although both rRTIN_55_ and rRTIN_212_ retained a certain ability to generate -sscDNA at 1 or 10 μM dNTPs, the Km values for dNTPs were significantly higher than those for rRTIN. Similarly, RTIN_55_ and rRTIN_212_ showed increased sensitivity to EFV with lower EC_50_ values compared to rRTIN (Appendix A). These results suggest that the full-length form of IN with three domains was necessary to induce sufficient allosteric effects on RT enzymatic functions.

### 3.6. MD Simulations of RT and RTIN Proteins

To gain structural insights into the impacts of IN fusion on the biochemical activities of RT, we conducted MD simulations of RT and RTIN proteins. Molecular models of RTIN fusion protein and RT protein of the HIV-1_NL4-3_ clone were constructed as described in Materials and Methods (Figure 6a) and subjected to MD simulations using Amber 16 [32]. The MD simulation is a computational method to delineate dynamic behaviors of biological macromolecules under thermal motions of atoms and molecular collisions in solution. This technique thus has been applied to characterize in silico the physical properties of biomolecules near the physiological conditions [38].

Structural dynamics during the simulations were evaluated with RMSD between the initial model structure and the structures at given time points. The RMSD of RT reached a near plateau immediately after the start of simulation, whereas that of RTIN reached a near plateau after 200 ns of the simulation (Figure 6b). These results indicate that the RTIN fusion protein requires more time to fold as a thermodynamically stable conformation and suggest that both RT and RTIN structures reached a state of thermodynamic equilibrium in solution conditions around 200 ns of MD simulations. These results are consistent with an analysis of the areas of exposed molecular surfaces of RT and RTIN in solution, where only the exposed surface of RTIN gradually decreased during MD simulations (Figure 6c).

### 3.7. Structural Impacts of IN Fusion on RT Active Center

The RT and RTIN structures at 500 ns of MD simulations were used to characterize the structural impacts of IN fusion on the RT active center (Figure 7a). Notably, the fingers and thumb subdomains of RT were positioned more closely in RTIN than RT via configurational changes in the thumb subdomain (Figure 7b). Consequently, noncovalent attractive interactions, such as hydrogen bonds, formed frequently between the thumb and fingers subdomains during MD simulations only with the RTIN fusion protein (Figure 7c). Such a hydrogen bond formation was not detected in the RT monomer lacking IN (Figure 7c, left panel). The intimate interactions between the thumb and fingers subdomains in RTIN were not permanent, and completely abolished once during 500 ns of MD simulations (Figure 7c, RTIN arrowhead). These results suggest that IN fusion into the C-terminal end of RT can allosterically influence the dynamic aspect of intramolecular interactions between the thumb and fingers subdomains of RT in *cis*.

Structural fluctuations of biological molecules in solution play crucial roles in molecular interactions [39,40,41] and thus in the biological phenotypes of viruses [28]. To gain insights into the possible impacts of IN fusion on fluctuations in the interaction surfaces of RT, we compared structural fluctuations of individual amino acid residues in the RT domain between RT and RTIN models by using RMSF as a quantitative indicator. The RMSF has been used to clarify *cis*-allosteric effects of genetic mutations [28,42,43,44] and effects of molecular interactions [30,31] on the physical properties of the interaction surfaces on the viral biomolecules. Basically, the magnitudes of the fluctuations were augmented in most of the amino acid residues in RTIN as compared with those in RT (Figure 8a, red line). Importantly, sites heavily influenced by IN fusion, with RMSF increases above 1.4 Å, contained 113D and 114A residues for dNTP binding [45] and NRTI binding [46] on the palm subdomain, and contained the β7/β8 loop on the fingers subdomain (136N, 137N, 138E, 139T, 140P and 141G) (Figure 8a). Fluctuations in NNRTI binding pockets on the palm subdomain [46,47] were also augmented with an RMSF increase of 0.8-1.0 Å (Figure 8a). On the other hand, changes in the RMSF of RNA template binding regions, composed of the fingers, thumb, and palm subdomains, were mild and increased between 0.1–0.4 Å (Figure 8b). Figure 8b summarizes the three-dimensional locations of the influenced sites with an RMSF increase above 1.4 Å (red circle) along with those of amino acid residues involved in binding to natural substrate and inhibitors. These results suggest that IN-fusion into RT can influence the structural dynamics of the polymerization active center and inhibitor binding sites of RT in *cis*. 

## 4. Discussion

Our previous findings suggested that HIV-1 RT and IN need to be supplied in the RTIN fusion form for efficient reverse transcription to occur within cells [20], which in turn suggested the critical role of IN for HIV-1 reverse transcription through the RTIN form. In the present study, we demonstrated the impacts of IN fusion on RT activities and sensitivity to RT inhibitors, and provided the first evidence of the *cis*-regulation of IN, which occurs most probably by inducing a conformational change near RT catalytic sites. 

Using in vitro reverse transcription assay, we demonstrated that IN fusion at the C-terminal end of RT could markedly alter the biochemical properties of RT, i.e., the dNTP concentration dependence of reverse transcription (Figure 4) and the sensitivity to RT inhibitors targeting the polymerization active center (Figure 5). These results strongly suggest that IN fusion can regulate the biochemical phenotype of HIV-1 RT. Previous studies have suggested the presence of *trans*-acting roles of IN in RT enzymatic functions by stimulating RT processivity [49,50]. Although impact of RTIN on RT processivity remains to be determined, this is the first evidence that HIV-1 IN can regulate RT activity in *cis*. 

Concentrations of dNTPs vary depending on cell type and proliferation status [51]. For human primary T cells, major targets of HIV-1 infection, the concentration of dNTPs was estimated to be 1–5 μM in activated status [52]. In monocyte-derived macrophages, another target of HIV-1 infection, a much lower concentration was estimated. HIV-1 infection depends on the activation status of T cells [3] or macrophages [4]. Comparative analysis of rRTIN, rRTs, and protease-cleaved rRTIN revealed that the fusion of IN to RT significantly stimulated cDNA generation at low concentrations of dNTPs, i.e., less than 10 μM, which corresponds to the concentrations in activated T cells. Therefore, the superior RDDP activity of rRTIN at lower concentrations of dNTPs might have physiological significance for reverse transcription in the context of HIV-1 infection. Analysis of domain deletion mutants of RTIN showed that all three domains of IN are required for the full allosteric effect on RT (Appendix A). We noticed, however, that cDNA synthesis at low concentrations of dNTPs was partly retained in rRTIN55 or rRTIN212, which, respectively, contain only IN NTD or NTD-CCD. 

Our in vitro experimental data indicate that IN-fusion on RT can remotely influence the structural properties of the RT active center consisting of binding pockets to dNTP and RT inhibitors. To address this issue, we constructed RT and RTIN monomer models using AlphaFold2 [23] and conducted MD simulations. The in silico studies indeed revealed two types of marked structural impacts on the RT active center via IN fusion on RT: configurational and dynamics effects. First, our MD simulations suggested that IN fusion into RT can alter the configuration of the RT thumb subdomain, leading to the creation of weak de novo interactions between the thumb and fingers subdomains (Figure 7). In combination with the palm subdomain, these subdomains constitute a polymerase active center and participate in binding to the RNA template, dNTP and NRTI. Therefore, it is possible that the IN-induced conformational changes in these subdomains primarily alter the selectivity of substrate/NRTI and the efficiency of RT translocation, and thereby eventually lead to changes in its polymerization activity and/or in the NRTI susceptibility of RT. Our MD simulations also suggest that IN fusion can alter the fluctuation profiles of amino acid residues involved in binding to dNTP and NNRTI (Figure 8). It should be noted that the structural fluctuations of the interaction surfaces of biomolecules in solution are critical for the molecular interactions and thus for the overall function of the biomolecules [28,30,31,39,40,41,43,53,54]. Therefore, as indicated in this study, it is possible that the alterations in the magnitudes of fluctuations in the RT active center influence the binding efficiencies of the substrate and NNRTI to RT and contribute to changes in the biochemical phenotype of RT.

Our MD simulations suggest that IN fusion to RT induced the augmentation of amino acid fluctuations at both the dNTP and NNRTI binding sites of RT (Figure 8). This seems puzzling because IN fusion was likely to induce the opposite effects on the polymerization and NNRTI sensitivities of RT (Figure 4 and Figure 5). These data suggest that changes in fluctuations at the substrate/inhibitor binding sites alone are insufficient to induce IN fusion effects, and may imply that coordinated changes in other structural properties are necessary. Further study is necessary to clarify how the above two and/or other undescribed structural impacts coordinately regulate biochemical phenotypes of RT. Previous studies have shown that the interaction of HIV-1 RT and IN involves RT residue Ile178 [55,56]. However, at least in our MD analyses, we did not detect possible interaction of RT residue Ile178 with IN. This discrepancy might be explained by different effects of IN through cis and trans modes on RT structures. In addition, it remains to be clarified whether IN fusion can influence the RT structure in the context of an RTIN multimer, which might be the functional entity for the polymerization in vivo. What is clear at present is that IN fusion into the C-terminal end of the HIV-1 RT p66 subunit can allosterically regulate the structural properties of the polymerization active center and inhibitor binding site at the monomer level.

Within the HIV-1 virus particle, some portions of RT and IN were kept in fused (RTIN) form, although the bulk of the Pol precursor was cleaved into individual forms by viral protease [20]. Indeed, rRTIN was proteolyzed by HIV-1 protease quite efficiently (Figure 1b). Currently, the underlying mechanism that keeps the RTIN form by avoiding complete proteolysis by HIV-1 protease remains to be elucidated. One intriguing possibility is that some portion of intermediate products of the Pol precursor might associate with vRNA, which makes the Pol precursor resistant to cleavage by HIV-1 protease. In some retroviruses (α-retrovirus), RT enzyme is composed of a heterodimer of RT (α) and incompletely processed RTIN (β) subunits in the avian leukosis viruses [57]. Interestingly, the β subunit of the α-retrovirus RT has many allosteric properties, including specific binding to tRNA^trp^ primer and controlling IN-mediated DNA endonuclease activity [58]. Our present data might reflect allosteric regulation of HIV-1 RT by IN, which has been noticed for the β subunit of the α-retrovirus RT.

Although our present data demonstrated that RTIN has enzymatic superiority over RT in RDDP activity in low dNTPs, the cleaved form of RT (p66/p51) might have critical roles in completing reverse transcription where higher concentrations of dNTPs are available, such as in nuclear compartments. Since integration catalyzed by IN is also a critical step to retrovirus infection following reverse transcription of a retrovirus genome, IN enzymatic activity in RTIN form is another important issue to address. However, we could not detect any significant integrase activity of rRTIN. In this regard, it seems likely that IN that is already cleaved in a virus particle might play a main role in integration. Alternatively, RTIN might be cleaved into IN by cellular protease during the early events of HIV-1 replication. We favor the latter possibility, since our previous study showed that *trans*-complementation of RT and IN separately did not rescue HIV-1 lacking RT and IN [20], suggesting that retroviral reverse transcription and integration might be tightly connected to each other. Recent studies using imaging and biochemical assays to track HIV-1 capsids and virus gRNA suggested that uncoating and reverse transcription may occur in nuclear compartments [59,60,61]. These studies led us to revisit our current understanding of HIV-1 early replication events [62]. Reverse transcription and integration might be connected to each other in a heretofore unknown manner. The RTIN functions elucidated in the present study would serve as a platform upon which to understand HIV-1 early replication events and develop a novel anti-retrovirus strategy.

## Figures and Tables

**Figure 1 viruses-15-00031-f001:**
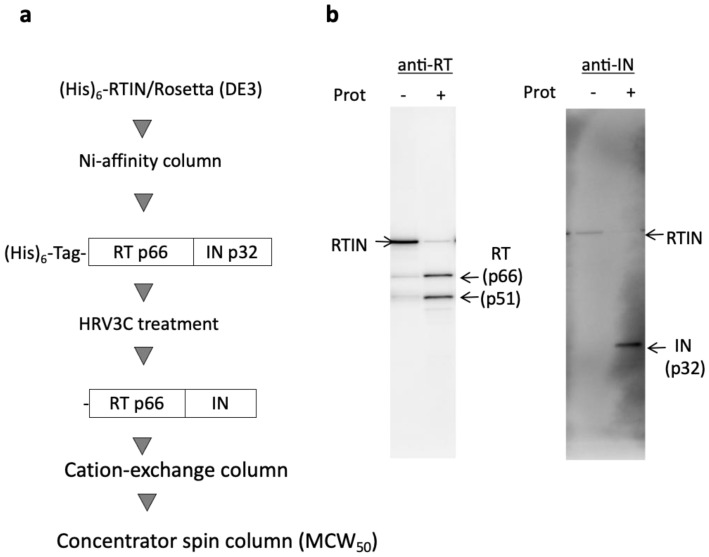
Preparation of recombinant HIV-1 RTIN protein. (**a**) (His)_6_-tagged form of HIV-1 RTIN was expressed in *E. coli* Rosetta (DE3) by incubation with 0.1 mM IPTG for 20 hrs at 18 °C. (His)_6_-tagged RTIN in the soluble fraction was purified through a Ni-affinity column. The (His)_6_-tag was then removed by treatment with HRV3C protease followed by glutathione-sepharose column chromatography to remove the HRV3C. The flow-through fraction was then subjected to cation exchange chromatography. Fractions eluted by 1×SP buffer [20 mM HEPES-NaOH (pH 7.0), 0.1 mM EDTA, 0.01% TritonX-100, 10% glycerol, and 1 mM DTT] containing 250–500 mM NaCl were pooled. The purified recombinant RTIN (rRTIN) was concentrated through a concentrator spin column with a filtration cutoff of 50 kDa (MCW_50_) and stored in 1×SP buffer containing 300 mM NaCl at −80 °C. (**b**). Purified rRTIN with (+) or without (−) HIV-1 protease treatment was subjected to Western blot analysis using an anti-RT (left) or anti-IN (right) antibody.

**Figure 2 viruses-15-00031-f002:**
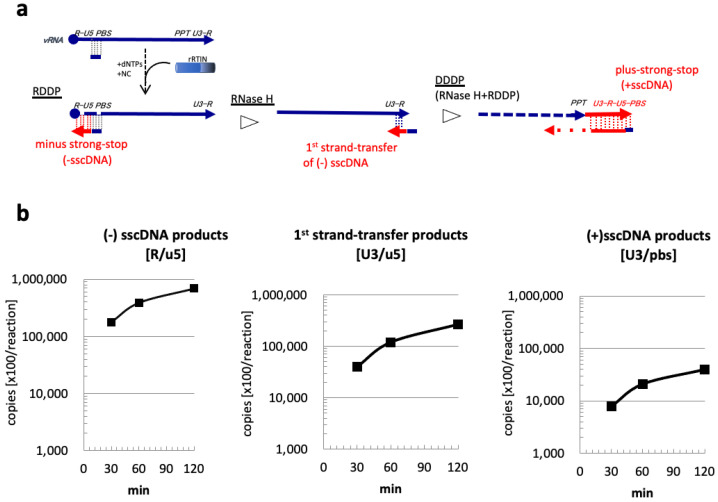
In vitro RT assay using rRTIN. (**a**) rRTIN was subjected to in vitro reverse transcription assay as described previously [22]. cDNAs of minus strong stop (−sscDNA), strand-transfer of -sscDNA (1st strand-transfer) products, and plus strong stop (+sscDNA), generated by RNA-dependent DNA polymerase (RDDP), RNase H and DNA-dependent DNA polymerase (DDDP) activities, are schematically depicted. (**b**). At 30, 60, or 120 min after reaction, the levels of cDNA intermediates were determined by using the primer set for the R/u5, U3/u5, or U3/pbs region to monitor RDDP, RNaseH and DDDP activities. Each value shown is a copy number of a cDNA in 1 μL of the reaction mixture at each time point.

**Figure 3 viruses-15-00031-f003:**
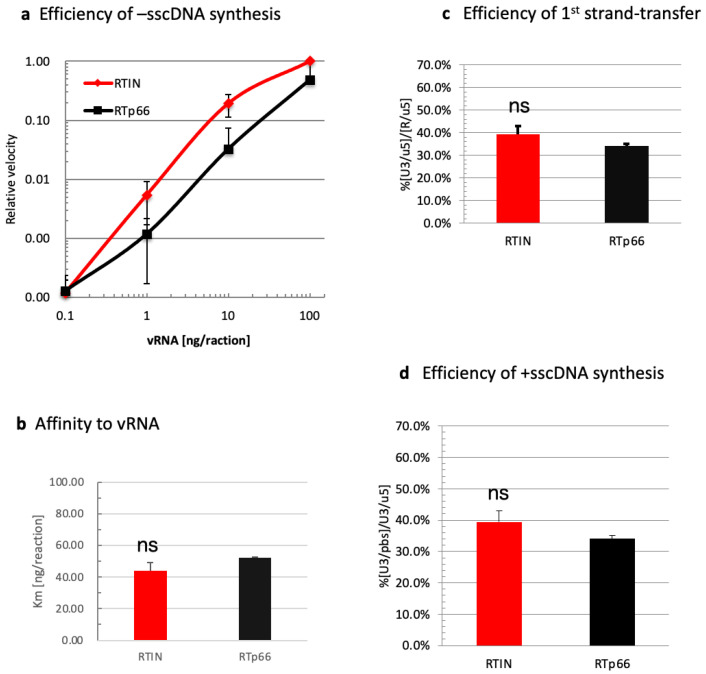
Impact of IN fusion on affinity of RT with vRNA. (**a**) An in vitro reverse transcription assay was performed in the presence of different doses of vRNA (0.1, 1, 10, 100 ng). The R/u5 copy number generated during 60 min incubation was measured and the value at each dose was plotted as a relative velocity to the reaction of rRTIN with 100 ng of vRNA as 1.0. The value at each vRNA dose is shown as means ± SD (*n* = 3). (**b**) Km for vRNA was estimated by calculating velocity at vRNA (100 ng/reaction) as the maximum velocity in the assay condition. Student’s *t* analysis showed no significant differences between RTIN and RTp66 (ns, *p* > 0.05). (**c**) The efficiency of the 1st strand-transfer was determined by calculating the relative (%) amount of the U3/u5 to that of each R/u5 product (% of [U3/u5]/[R/u5]) in the reaction with 100 ng vRNA). d. The efficiency of the +sscDNA synthesis was determined by calculating the relative (%) amount of the U3/pbs to that of each U3/u5 product (% of [U3/pbs]/[U3/u5]) in the reaction with 100 ng vRNA. (**c**,**d**) Student’s *t* analysis showed no significant differences (ns) between RTIN and RTp66 (*n* = 3, *p* > 0.05).

**Figure 4 viruses-15-00031-f004:**
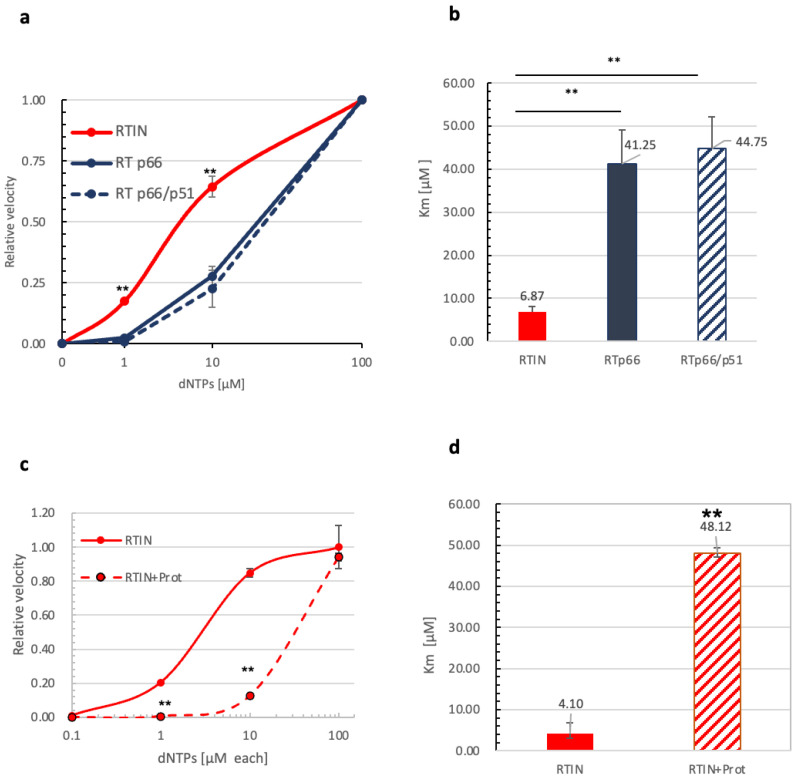
Impact of IN fusion on affinity of RT to dNTPs. (**a**) A reaction was performed in the presence of different doses of dNTPs (0.1, 1, 10, 100 μM) with a fixed vRNA dose (100 ng/reaction) and RTIN or RTs. The R/u5 copy number generated during 60 min incubation was measured, and the value at each dose was plotted as a velocity relative to the reaction with 100 μM of dNTPs as 1.0. The value at each dNTP dose is shown as means ± SD (n = 3). (**b**) The Km for dNTPs was estimated by calculating velocity at 100 μM dNTPs as the maximum velocity in the assay condition. The Km value of each protein is indicated on the top of each bar. Student’s *t* analysis showed significant differences between RTIN and either RTp66 or RTp66/p51 (** *p* > 0.01, *n* = 4). (**c**) Effect of protease cleavage of RTIN. rRTIN with or without digestion with HIV-1 protease was subjected to reaction, and data are plotted as described for a. (**d**) The Km for dNTPs was estimated by calculating velocity at 100 μM dNTPs as the maximum of velocity in the assay condition. The Km value of each protein is indicated on the top of each bar. Student’s *t* analysis showed significant differences between rRTIN with and without protease treatment (** *p* > 0.01, n = 4).

**Figure 5 viruses-15-00031-f005:**
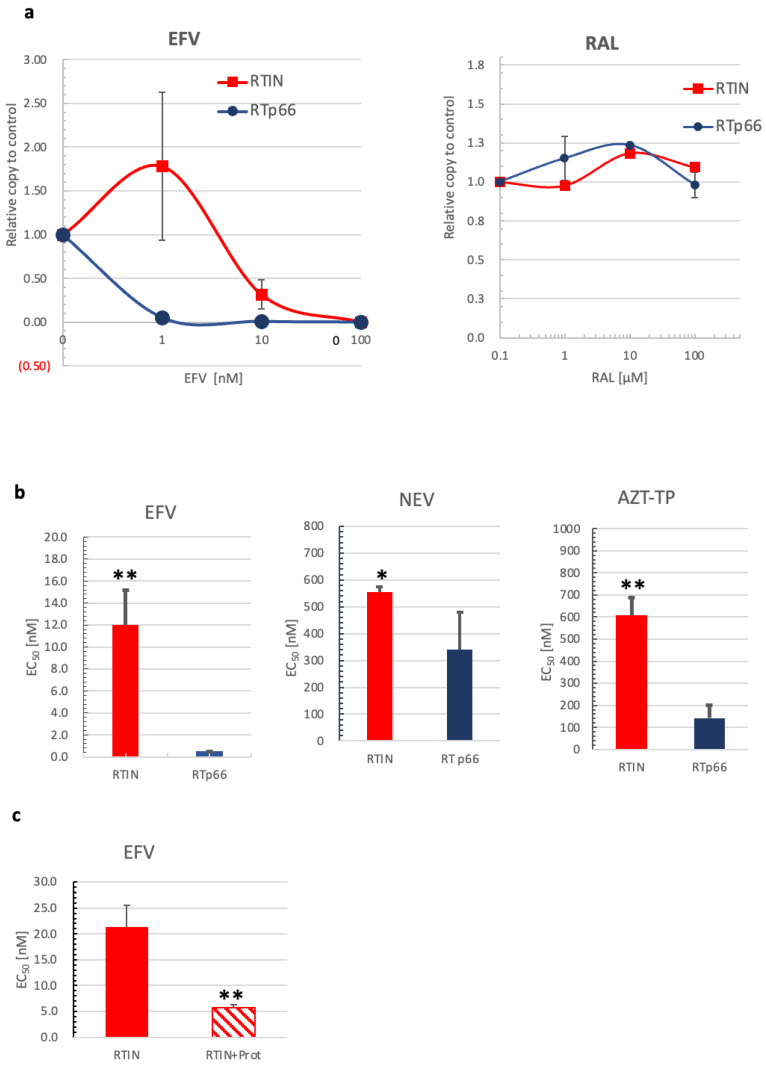
Effect of IN fusion on sensitivity to RT inhibitors. (**a**) An in vitro reverse transcription assay was performed in the presence of different concentrations of efavirenz (EFV) or raltegravir (RAL). Levels of R/u5 products were measured and are shown as values relative to the level with solvent (DMSO) only as 1.0. (**b**) The effective concentration 50 (EC_50_) of EFV, nevirapine (NEV), or 3‘-Azido-2′,3‘-dideoxythymidine-5′-triphosphate (AZT-TP) for RTIN or RTp66 was determined by calculating the drug concentration to inhibit 50% of RDDP activity relative to control. Student’s *t* analysis showed significant differences between rRTIN and RTp66 (** *p* > 0.01, * *p* > 0.05, *n* = 3). (**c**) The effect of cleavage by HIV-1 protease on the EFV sensitivity of RTIN was examined as described in a. The EC_50_ of EFV for rRTIN with or without pretreatment with HIV-1 protease is shown. Student’s *t* analysis showed that rRTIN had significant effect on EC_50_ by pretreatment with HIV-1 protease (** *p* > 0.01, *n* = 3).

**Figure 6 viruses-15-00031-f006:**
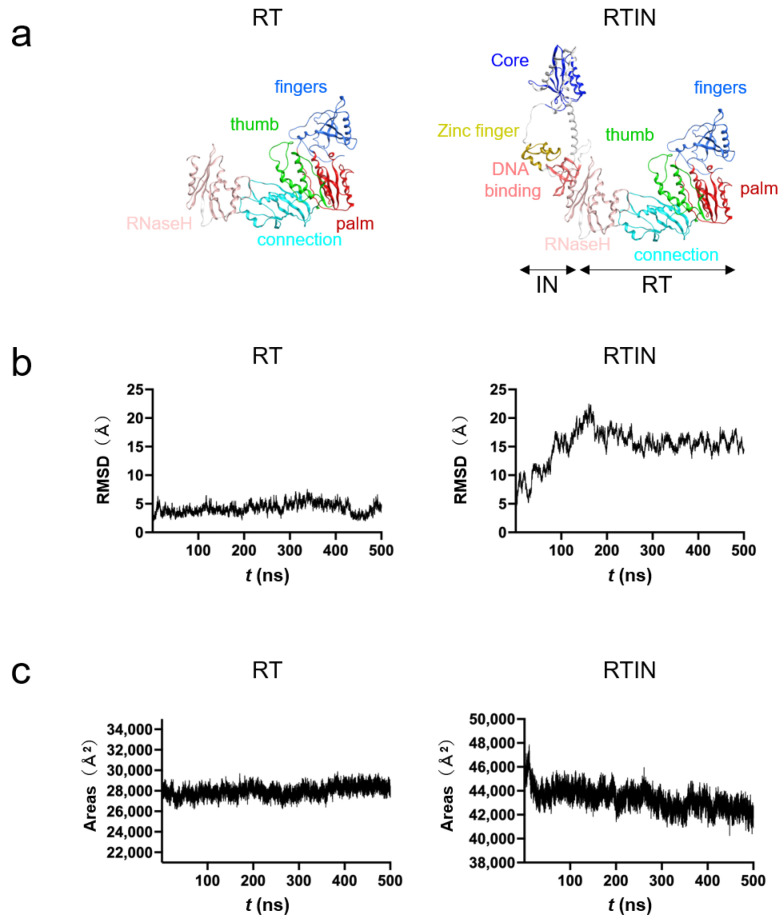
MD simulations of RT and RTIN proteins of the HIV-1_NL4-3_ clone. The amino acid sequence of the RTIN portion of the Pol region of the HIV-1_NL4-3_ molecular clone [24] was used to construct an RTIN model by the AlphaFold2 program [23]. The obtained RTIN model gave a per-residue confidence score (pLDDT) of 83.3, in the “Confident” range [23]. An RT model was constructed with the RT domain of the RTIN model by deleting the IN domain using MOE (Chemical Computing Group, Montreal, Quebec, Canada). The RT and RTIN models were subjected to MD simulations using modules in Amber 16 [32] as described previously [28,29,31]. (**a**) Initial models of RT and RTIN fusion protein before MD simulations. (**b**) Structural dynamics of RT and RTIN fusion protein during MD simulations. RMSDs between the structure of the initial model and those at given time points of MD simulation were used to monitor the overall structural changes during simulations. (**c**) Analysis of areas of exposed surfaces of RT and RTIN in solution during MD simulations.

**Figure 7 viruses-15-00031-f007:**
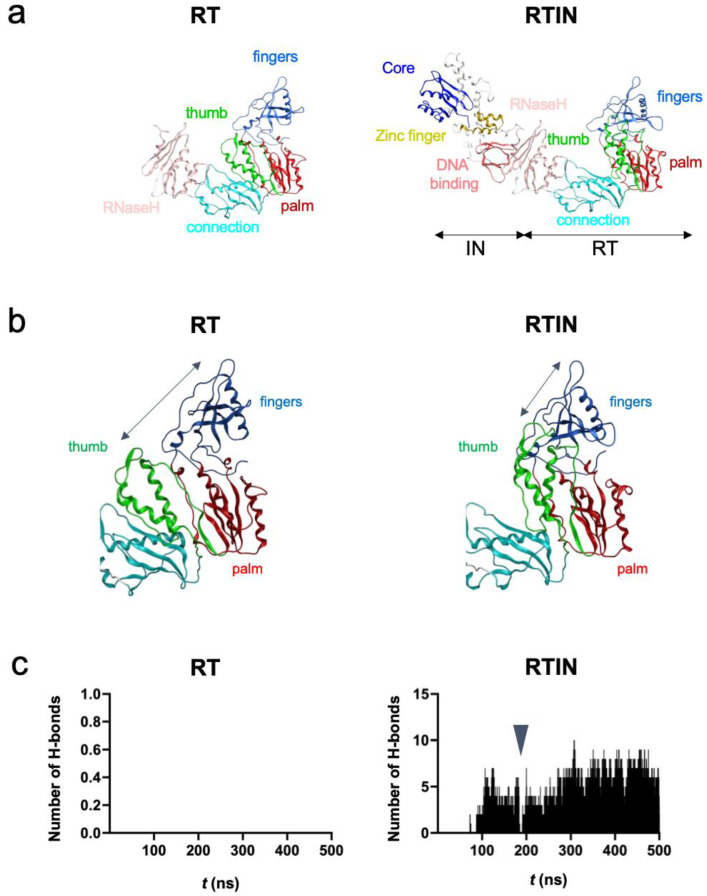
Effects of IN fusion on conformation of RT protein. Structures obtained from the MD simulations in Figure 6 were used to characterize impacts of RTIN fusion on RT. (**a**) Comparison between overall structures of RT and RTIN fusion protein at 500 ns of MD simulations. (**b**) Enlarged view of active center of RT in RT and RTIN fusion protein at 500 ns of MD simulations. (**c**) Numbers of hydrogen bonds formed between the thumb and fingers subdomains of RT during 500 ns of MD simulations. The trajectory files during 500 ns of MD simulations were used to calculate the number of hydrogen bonds between the fingers and thumb subdomains in the RT region using the *cpptraj* module in AmberTools 16 [32].

**Figure 8 viruses-15-00031-f008:**
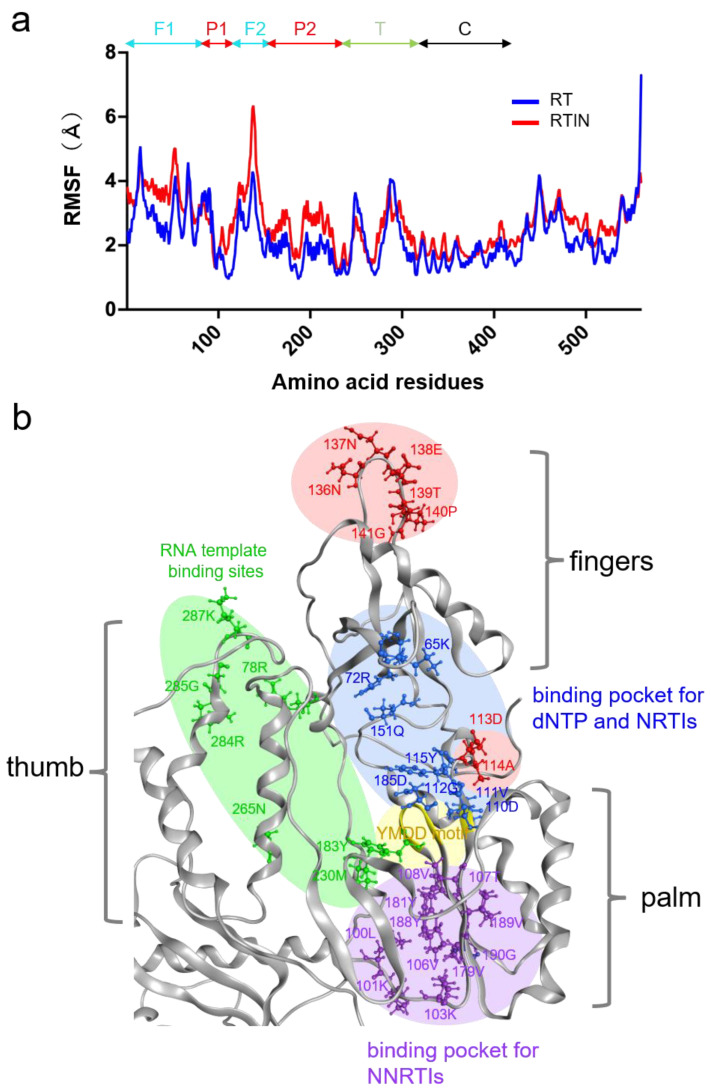
Effects of IN fusion on fluctuations of RT protein. RMSF values, which indicate the atomic fluctuations of the main chains of individual amino acids during MD simulations, were calculated using 50,000 snapshots in the equilibrium states during the last 100 ns of MD simulations using the *cpptraj* module in AmberTools 16, a trajectory analysis tool [32]. (**a**) Distributions of RMSF in RT. Numbers on the horizontal axes indicate positions in the mature RT of HIV-1 NL4-3 (GenBank accession no. AAK08484) [24]. P1 and P2, palm subdomain; F1 and F2, fingers subdomain; T, thumb subdomain; C, connection subdomain. (**b**) Overall view of RT active center. The red shaded areas indicate regions where RMSF increases to above 1.4 Å by IN fusion. Light yellow; YMDD motif of polymerase active center [45,48]. Light blue; region involved in dNTP binding [45]. Light green; region of RNA template binding [45]. Light purple; region of NNRTI resistance [45].

## Data Availability

Please contact the corresponding authors (T.M. and O.K.) for data on the present study.

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
