# Peer review of "Cis-Allosteric Regulation of HIV-1 Reverse Transcriptase by Integrase"

_viruses, 2022, doi:10.3390/v15010031_

Round 1

Reviewer 1 Report

In this very interesting in vitro study, Masuda et al report that RT and IN in fusion form (RTIN) (uncleaved) in contrast to rRT generated higher cDNAs under physiological concentrations of dNTPs, suggesting an increased affinity of RTIN to dNTPs. Using a diverse panel of assays, they showed for the first time the cis-regulation of IN on RT activity and sensitivity to RT inhibitors.

Overall, these data will be helpful to eventually figure out the role of RTIN in vivo.

Minor comments:

1. Fig. 1b. Why there are faint bands for p66 and p51 in the sample (-) Prot in the anti-RT blot?

2. Fig. 2b. The q-PCR products using rRTIN should be compared in the same experiment with q-PCR using  rRT. Based on ref 22 it is difficult to estimate the efficiency of rRTIN. At a first approximation it looks like there is at least 1 log lower in q-PCR products using rRTIN compared to rRT  despite using 0.1 pmole of vRNA compared to ref 22 that used maximum 83 fmol.

3. Fig. 3c and fig. 3d. Please show the statistics.

Author Response

  1. 1b. Why there are faint bands for p66 and p51 in the sample (-) Prot in the anti-RT blot?

Authors’ Response: The bands for p66 and p51 were diversely generated depending on amount of IPTG and temperature for the induction of RTIN in culture. The faint bands for p66 and p51 detected in rRTIN preparation (Figure 1b, Prot -), therefore, might be generated most probably by E. coli-derived protease(s) during induction phase of rRTIN. This description was added in the text (lines 181-183 in revised manuscript).

  1. 2b. The q-PCR products using rRTIN should be compared in the same experiment with q-PCR using  rRT. Based on ref 22 it is difficult to estimate the efficiency of rRTIN. At a first approximation it looks like there is at least 1 log lower in q-PCR products using rRTIN compared to rRT  despite using 0.1 pmole of vRNA compared to ref 22 that used maximum 83 fmol.

Authors’ Response: Comparative analysis between RTIN and RT was presented in the following experiments in detail under different conditions (Fig. 3 and Fig.4). Since Fig. 2b was shown to demonstrate possession of RT enzymatic activities of rRTIN, we modified the last sentence by deleting phrase of “that were almost equivalent to those of RTp66 under these experimental conditions” in the revised manuscript (line 209 in revised manuscript).

  1. 3c and fig. 3d. Please show the statistics.

Authors’ Response: Student’s t analysis showed no significant differences between RTIN and RTp66 (n=3, ns, p>0.05).  We added this result of statistics analysis for Fig. 3c and d and Figure legend (lines 247-248 in revised manuscript).

Reviewer 2 Report

Cis-allosteric regulation of HIV-1 RT by IN

Masuda et al. present a very interesting paper on cis-regulatory properties of the HIV-1 IN protein fused to the C-terminus of its RT protein.  This manuscript is a major follow-up to their previous published efforts to characterize different reverse transcriptase constructs during virus infection of cells. 

The preparation of recombinant HIV-1 RTIN appears straightforward and correct with respect to their immunological studies identifying the p66 and p51 proteins and RTIN (p66 and IN fused together), and its cleavage by HIV-1 protease of RTIN to produce p66, p51 and IN. The purified RTIN exists as an aggregate (SEC profile) but appears to display all of the correct viral DNA products using an RNA pbs-primer and U5 vRNA fragment (R/U5) as well as U3/U5 and U3/pbs substrates. PCR amplification of these DNA products showing the correct synthesis timeline, paralleling in vivo reactions, is very convincing. Is the need to run the RT reaction at 42 C necessary to increase the solubility of the recombinant RTIN?  What kind of results for viral DNA synthesis occurs at 37 C, the normal temperature for reverse transcription? Some comparison data at both temperatures would be informative.

The authors demonstrate RTIN has a slightly higher velocity than RTp66 (single subunit that contains RT and RNAaseH activities) for -sscDNA and +sscDNA. The data appears correct. In Figure 4, the authors demonstrate the RTIN possesses a significantly higher affinity the RT p66 or RT p66/61. Cleverly, addition of protease with RTIN demonstrated the same result, that is, RTIN has a higher velocity at the lower concentrations of dNTPs.  Another interruption maybe that RTIN possess higher RT processivity (Dobard, et al.,2007), thus the higher velocity, than the other RT species. Can the authors rule out this possibility? Maybe it is both.

The differential sensitivity of RTIN and RTp66 to RT inhibitors, with and without protease cleavage, is interesting (Fig. 5). The IN fused fragment has a major effect on the ability of EFV to inhibit reverse transcription suggesting that IN significantly influences RT functions.

This reviewer is not familiar with MD simulations. How does the AlphaFold 2 program derived structure of RT p66 or RTIN compare to the many cryo-EM and x-ray structure of p66 or p66/p51?  Some discussion would appear appropriate.

A recent review in Viruses (Multimodal Functionalities of HIV-1 Integrase) stated that the β subunit of the α-retrovirus αβ reverse transcriptase has many of the allosteric properties as HIV RTIN.  The αβ polymerase consists of the β subunit that contains the p32 IN moiety. The α polymerase (p66) possesses RT and RNaseH activity. The β subunit binds specifically to tRNAtrp  while the α subunit polymerase does not, possibly helping to explain the results by Masuda et al. The αβ polymerase has significantly processivity for both its RNaseH and polymerase activities than the single subunit α polymerase. In addition, the α moiety in the β subunit silences the Mg++ dependent DNA endonuclease activity that is associated with the IN moiety until it is release by protease activity. These past data further support this Masuda et al. manuscript that these regions display cis-allosteric regulation properties.   Some discussion of the α-retrovirus system appears appropriate for the readers not familiar with reverse transcriptase.

Author Response

1) Is the need to run the RT reaction at 42 C necessary to increase the solubility of the recombinant RTIN?  What kind of results for viral DNA synthesis occurs at 37 C, the normal temperature for reverse transcription? Some comparison data at both temperatures would be informative.

Authors’ Response: We think results might not differ between 37°C and 42°C, since both of RT and RTIN generated higher cDNA at 42°C compared to 37°C. The RT reaction at 42°C was applied to avoid mis-annealing of vRNA and to facilitate efficient and correct 1st strand-transfer as described before [22]. Brief description for the reaction temperature at 42°C was added in Materials and Methods (lines 111-112 in revised manuscript).

 2. Another interruption maybe that RTIN possess higher RT processivity (Dobard, et al.,2007), thus the higher velocity, than the other RT species. Can the authors rule out this possibility? Maybe it is both.

Authors’ Response: Impact of RTIN on RT processivity was not directly addressed in this study. So, at this stage, we cannot rule out the possible impact of RTIN on the RT processivity. As suggested by reviewer #2, we also think that both might be involved. With consideration of this point, we modified text in discussion to state that impact of RTIN on RT processivity remains to be determined (lines 429-432 in revised manuscript).

3. How does the AlphaFold 2 program derived structure of RT p66 or RTIN compare to the many cryo-EM and x-ray structure of p66 or p66/p51?  Some discussion would appear appropriate.

Authors’ Response: We thank Reviewer #2 for the valuable comments. We added a following sentence in Materials and Methods (lines 132-137 in revised manuscript). The accuracy of the RTIN model was evaluated by the Root-mean-square deviation (RMSD) score between the predicted RTIN model and the reported X-ray crystal structure of the RT or IN (PDB code: RT; 3HVT [25], IN; 1K6Y [26] and 1EX4 [27]) using the “Structure Superposition” tool in MOE. Each domain of RTIN model was similar to the reported RT or IN models (RMSD score: RT; 2.161 Å, IN-NTD; 1.931 Å, IN-CCD; 1.81 Å, IN-CTD; 0.785 Å).

4. Some discussion of the α-retrovirus system appears appropriate for the readers not familiar with reverse transcriptase.

Authors’ Response: We thank for this reviewer for this important suggestion. We added following discussion in the revised manuscript. In some retroviruses (α-retrovirus), RT enzyme is composed of a heterodimer of RT (α) and an incompletely processed RTIN (β) subunits in the avian leukosis viruses [57]. Interestingly, the β subunit of the α-retrovirus RT has many of the allosteric properties including specific binding to tRNAtrp primer and controlling IN-mediated DNA endonuclease activity [58]. Our present data might reflect allosteric regulation of HIV-1 RT by IN, which has been noticed for the β subunit of the α-retrovirus RT (lines 491-497 in revised manuscript).

Reviewer 3 Report

The paper is fine as it is.  The methodology carried out is scientifically sound. It is well written and the findings are based on the experiments carried out. 

Author Response

We thank Reviewer #3 for careful reading and positive evaluation of our manuscript.

Reviewer 4 Report

A very interesting original article devoted tothe study of the mechanisms of regulation of the expression of proteins of the pol gene of the HIV. The authors of the article for the first time managed to demonstrate the regulatory role of HIV integrase in the structural organization and enzymatic activity of reverse transcriptase. The article will arouse great interest among virologists dealing with the problem of HIV infection  and structural organization of viral proteins. The article can be published without changes.

Author Response

We thank Reviewer #4 for careful reading and positive evaluation of our manuscript.